# Automated Design of Metaheuristic Algorithms: A Survey

**Qi Zhao**[1]                                                                    *zhaoq@sustech.edu.cn*

**Qiqi Duan**[2]                                                          *11749325@mail.sustech.edu.cn*

**Bai Yan**[1]                                                                     *yanb@sustech.edu.cn*

**Shi Cheng** [3]                                                               *cheng@snnu.edu.cn*

**Yuhui Shi**[1,†]                                                             *shiyh@sustech.edu.cn*
[1] *Southern University of Science and Technology, China*
[2] *Harbin Institute of Technology, China*
[3] *Shaanxi Normal University, China*

**Reviewed on OpenReview:** *https://openreview.net/forum?id=qhtHsvF5zj*

## Abstract

Metaheuristics have gained great success in academia and practice because their search logic can be applied to any problem with an available solution representation, solution quality evaluation, and notion of locality. Manually designing metaheuristic algorithms for solving a target problem is criticized for being laborious, error-prone, and requiring intensive specialized knowledge. This gives rise to increasing interest in automated design of metaheuristic algorithms. With computing power to fully explore potential design choices, the automated design could reach and even surpass human-level design and could make high-performance algorithms accessible to a much wider range of researchers and practitioners. This paper presents a broad picture of automated design of metaheuristic algorithms, by conducting a survey on the common grounds and representative techniques in terms of design space, design strategies, performance evaluation strategies, and target problems in this field.

## 1 Introduction

Metaheuristic algorithms are stochastic search methods that integrate local improvement with high-level strategies of escaping from local optima (Glover and Kochenberger, 2006). Representatives include genetic algorithm (GA) (Holland, 1973), simulated annealing (SA) (Kirkpatrick et al., 1983), tabu search (Glover, 1989), particle swarm optimization (PSO) (Kennedy and Eberhart, 1995), ant colony optimization (ACO) (Dorigo et al., 1996), and memetic algorithms (Moscato, 1999), etc. In contrast to analytical methods and problem-specific heuristics, metaheuristics can conduct search on any problem with available solution representation, solution quality evaluation, and a certain notion of locality, in which locality denotes the ability to generate neighboring solutions via a heuristically-informed function of one or more incumbent solutions (Swan et al., 2022). Such capability has enabled metaheuristics to gain broad success across various fields.

Usually, human experts are requested to manually tailor algorithms to a target problem to obtain good enough solutions. Although many manually tailored algorithms have been reported to perform remarkably on various problems, manual tailoring suffers from apparent limitations. First, the manual tailoring process could be laborious, which may cost the expert days or weeks conceiving, building up, and verifying the algorithms. Second, manual tailoring could be error-prone due to the high complexity of the target problem

---

[†]Corresponding author.

and the high degree of freedom in tailoring the algorithm. Third, the manual process is untraceable regarding what motivates certain design decisions, losing insights and principles for future reuse (Swan et al., 2022). Lastly, manual tailoring is unavailable in scenarios with an absence of human experts.

The automated design is a promising alternative to manual tailoring to address the limitations. It leverages today's increasing computing resources to create metaheuristic algorithms to fit for solving a target problem. Herein, the computer (partly) replaces human experts to conceive, build up, and verify design choices. Human experts can be involved but are not necessary during the process. Therefore, the automated design could make high-performance algorithms accessible to a much broader range of researchers and practitioners. This is significant in response to the lack of time and labor resources. Furthermore, by leveraging computing power to fully explore potential design choices, the automated design could be expected to reach or even surpass human-level design. In the long run, it could be a critical tool in the pursuit of autonomous and general artificial intelligence.

**Related work:** By automatically tailoring algorithm components, structures, and hyperparameter values, the automated design could find either instantiations/variants of existing algorithms or unseen algorithms with novel structures and component compositions (Stützle and López-Ibáñez, 2019; Qu et al., 2020). In principle, the automated design can be conducted either offline by a target distribution of problem instances or online by the search trajectory (Swan et al., 2022). Most automated design works follow the offline manner, which is significant for scenarios where one can afford a priori computational resources (for design) to subsequently solve many problem instances drawn from the target domain. We distinguish the relationship of automated design to related topics, i.e., automated algorithm selection (Kerschke et al., 2019), automated algorithm configuration (Huang et al., 2020; Schede et al., 2022), adaptive operator selection and hyperheuristics (Burke et al., 2013; Pillay and Qu, 2018), to avoid conceptual confusion:

- Automated algorithm selection (Kerschke et al., 2019) chooses algorithms from a portfolio for solving a specific problem or problem instance. It alleviates the limitations of manual algorithm design by the selection mechanism that allocates suitable algorithms to different problems or problem instances. The output of the selection is one of the existing algorithms, instead of a customized algorithm for the target problem. From this perspective, automated algorithm selection is an independent topic with respect to automated design.

- Automated algorithm configuration appears in two kinds of literature. The first kind is offline tuning or online controlling hyperparameters of an existing algorithm (Eiben et al., 1999; Huang et al., 2020). The output is an instantiation of the existing algorithm. The second kind involves both offline hyperparameter tuning and algorithm component composition via representing components as categorical parameters (Blot et al., 2017; Aydın et al., 2017; Blot et al., 2019; Sae-Dan et al., 2020; Tari et al., 2020a). Despite subjecting to given algorithm templates, the second kind performs the same as a part of automated design, i.e., component composition and parameter configuration. Thus, the second kind of automated algorithm configuration falls into automated design.

- Adaptive operator selection and hyperheuristics use high-level automated methods to select or generate low-level heuristics/metaheuristics (Burke et al., 2019). The high-level methods are usually heuristics/metaheuristics, e.g., genetic programming (GP) (Koza, 1994), as well as machine learning and data mining (e.g., exploratory landscape analysis) methods; the primary goal of hyperheuristics is raising the low-level heuristics' generality (Swan et al., 2017; Burke et al., 2019; Pillay and Qu, 2021). In comparison, methods for automated design include not only GP, but also those from other fields, e.g., from the hyperparameter optimization of automated machine learning (Hutter et al., 2019); goals of automated design include but are not limited to generality (Ye et al., 2022a). In this regard, the hyperheuristics that generate metaheuristic algorithms can be seen as a part of automated design of metaheuristic algorithms.

Many efforts have been devoted to the automated design of metaheuristic algorithms in recent years. There have been several surveys (Kerschke et al., 2019; Huang et al., 2020; Stützle and López-Ibáñez, 2019; Schede et al., 2022) relating to some of the efforts, but a comprehensive compilation is lacking. The surveys in Kerschke et al. (2019); Huang et al. (2020); Schede et al. (2022) are for automated algorithm selection or

configuration, the scopes of which differ from or only cover a part of automated design according to the above concept discrimination. The work in Stützle and López-Ibáñez (2019) is for automated design and focuses on the methods of design. However, apart from the methods, different types of design space, ways of algorithm representations, paradigms for searching design choices, and metrics for evaluating the designed algorithms are also essential for conducting automated design. A systematic survey of these essentials is necessary but still lacking.

**Contributions:** This paper presents a broad picture of automated design of metaheuristic algorithms, by conducting a survey on the common grounds and representative techniques in this field. In the survey, we first provide a taxonomy of automated design of metaheuristic algorithms by formalizing the design process into four modules, i.e., design space, design strategies, performance evaluation strategies, and target problems. Unlike related surveys organized from the perspectives of methods for algorithm configuration/design, our taxonomy involves important elements of automated design that have not been involved in related surveys, such as types of design space, algorithm representations, and applications. Moreover, our taxonomy provides a comprehensive understanding of the four main modules of the automated design process, which would allow interested readers to easily overview existing studies on the modules of interest and make comparisons. Then, we overview the common grounds and representative techniques with regard to the four modules, respectively. We further discuss the strengths, weaknesses, challenges, and usability of these techniques, which would give researchers a comprehensive understanding of the techniques and provide practitioners with guidance on choosing techniques for different algorithm design scenarios. Finally, we list some research trends in the field, which would promote the development of automated design techniques for serving and facilitating metaheuristic algorithms for complicated problem-solving.

The rest of the paper is structured as: Section 2 introduces preliminaries of this survey; Sections 3, 4, 5, and 6 review and discuss the research progress on the four main modules of automated design of metaheuristic algorithms, i.e., design space, design strategies, performance evaluation strategies, and target problems, respectively; Section 7 points out future research directions; finally, Section 8 concludes the paper.

## 2 Preliminaries

### 2.1 Formulation of Metaheuristic Algorithm Design

Given a target problem, through algorithm design, we would like to find an algorithm (or algorithms) to fit for solving the problem:

$$\arg\max_{A \in \mathcal{S}} \ \mathbb{E}_{\mathcal{I}}\big[\mathbb{E}_{\mathcal{P}}[P(A|i)]\big], \ i \in \mathcal{I}, P \in \mathcal{P}, \tag{1}$$

where $A$ is the algorithm to be designed; $\mathcal{S}$ is the design space, from where $A$ can be instantiated; $i$ is an instance from the target problem domain $\mathcal{I}$; $P : \mathcal{S} \times \mathcal{I} \rightarrow \mathbb{R}$ is a metric that scores the performance of $A$ by a run of $A$ on $i$. Because metaheuristic algorithms conduct stochastic search, we need to estimate the expected performance over $\mathcal{P}$, i.e., multiple runs of $A$ result in multiple $P \in \mathcal{P}$.

In reality, the distribution of problem instances in $\mathcal{I}$ is usually unknown; and one cannot exhaust all the instances during the design process. The common practice of settling this is to consider a finite set of instances from $\mathcal{I}$. Consequently, Eq. (1) is reformulated as

$$\arg\max_{A \in \mathcal{S}} \ \mathbb{E}_{I_t}\big[\mathbb{E}_{\mathcal{P}}[P(A|i)]\big], \ i \in I_t \subseteq \mathcal{I}, \forall t \in \{1, 2, \cdots, T\}, \ P \in \mathcal{P}, \tag{2}$$

where $I_t$ is the finite set of problem instances that are targeted at time (i.e., iteration[1]) $t$ of the design process. The instances can either be fixed (i.e., $I_1 = I_2 = \cdots = I_T$) or dynamically changed during the design process. The output of solving Eq. (2) is an algorithm (or algorithms) with the best performance on the instances. To avoid the designed algorithms overfitting, the design process should be followed by a test on the designed algorithms' generalization to instances from $\mathcal{I}\backslash\{I_1, I_2, \cdots, I_T\}$.

---

[1]Since Equation 2 is black-box, it is often solved in an iterative manner.

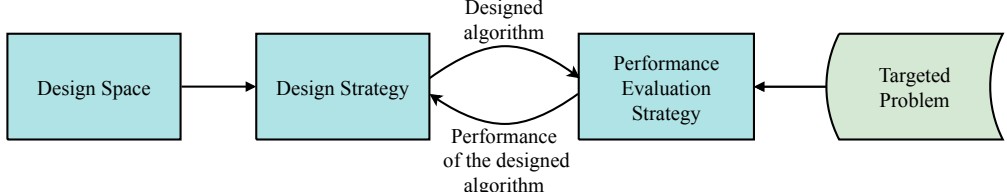

Figure 1: Abstractive process of automated design of metaheuristic algorithms.

## 2.2 Automated Design of Metaheuristic Algorithms

We abstract the general process of automated design of metaheuristic algorithms into four modules, as shown in Fig. 1. First, the design space collects candidate design choices (e.g., computational primitives, algorithmic operators, and hyperparameters), which regulates what algorithms can be designed in principle. Second, the design strategy provides a principle way to design algorithms by selecting and assembling design choices from the design space. Third, the performance evaluation strategy defines how to measure the performance of the designed algorithms. The measured performance guides the design strategy to search for desired algorithms. Finally, because the design aims to find algorithms to fit for solving a target problem, the target problem acts as external data to support the performance evaluation. The four modules constitute this survey's taxonomy of related research efforts.

## 2.3 Survey Scope

As stated in Section 1, the automated design, in principle, can be conducted either offline with a target distribution of problem instances or online with the search trajectory. Most automated design works follow the offline manner. In contrast, the online manner is often referred to in hyperheuristic or adaptive operator selection literature (related surveys can be found in Burke et al. (2013) and Pillay and Qu (2018)). The scope of this survey is the research efforts on offline automated design.

## 3 Design Space

The design space defines what metaheuristic algorithms can be found in principle. From the perspective of the elements within the design space, the current design space may be classified into two categories, namely, the design space with computational primitives and the design space with existing algorithmic operators. We first overview the research progress on the design space with computational primitives and discuss its strengths, weaknesses, and challenges in subsection 3.1. We then overview the design space with existing algorithmic operators, with a discussion on its strengths, weaknesses, and challenges in subsection 3.2. Finally, we analyze the usability of the two categories and their corresponding algorithm representations at the end of the section.

## 3.1 Design Space with Computational Primitives

Computational primitives define the elementary actions for computation. They include: arithmetic primitives, e.g., $+$, $-$, $*$, $/$; trigonometric primitives, e.g., $sin()$, $cos()$; probabilistic and statistic primitives, e.g., $max()$, $mean()$; operating instructions, e.g., $swap$, $duplicate$; etc. The design space consisting of these primitives enables designing metaheuristic algorithms by choosing primitives from the space, combining the chosen primitives as algorithmic operators, and composing the operators as an algorithm. The ways of representing algorithms over this design space include tree representation and linear array representation in the literature.

**Tree representation with computational primitives:** The binary GP-style tree (Koza, 1994) is the dominant representation for algorithms designed over computational primitives (Poli et al., 2005a;b; Burke et al., 2010; Vázquez-Rodríguez and Ochoa, 2011; Richter and Tauritz, 2018; Richter et al., 2019). As shown

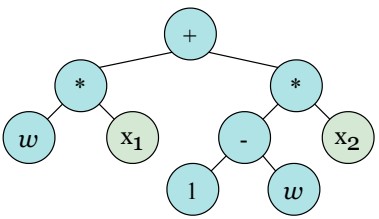

Figure 2: Example of a metaheuristic operator represented by a binary GP-style tree. This example represents the whole arithmetic crossover operator (Eiben et al., 2003), where $x_1$ and $x_2$ are the input variables (i.e., two parent solutions), $w$ is a weight parameter, and $+, *, -$ are primitives. The operator performs $w * x_1 + (1 - w) * x_2$ in mathematical form.

in Fig. 2, terminals of the tree are the inputs, including variables, parameters, and constants; non-terminals are the computational primitives. The operation specified by the computational primitive performs by using the values obtained from its child nodes; this process repeats until the root node is reached.

The tree was usually employed to represent a single algorithmic operator instead of an entire algorithm (Poli et al., 2005a;b; Burke et al., 2010; Vázquez-Rodríguez and Ochoa, 2011; Richter and Tauritz, 2018; Richter et al., 2019), because representing an entire algorithm leads to a large tree and could be infeasible to find an appropriate tree over the large search space (Woodward and Bai, 2009). An algorithm template is required to insert the operator represented by the tree to an algorithm. The template determines the algorithm structure, i.e., the execution order and logic of the involved operators. For example, in Poli et al. (2005a) and Poli et al. (2005b), the particle update operator was represented by a tree and designed; then, the designed operator was inserted into the PSO template (i.e., a recursive process of particle velocity update, particle location update, and best particle update) (Kennedy and Eberhart, 1995). Other examples include representing the mating selection operator of GA (Richter and Tauritz, 2018) and the individual selection operator for mean-update in CMA-ES (Richter et al., 2019).

Due to GP tree's expressiveness in representing programs over the primitives, the tree representation has been widely incorporated into the design space with computational primitives. An open challenge of using the tree representation is avoiding underfitting and overfitting. Current GP-based automated design literature presets the tree's size or structure (Richter et al., 2019; Nguyen et al., 2022), which can be seen as a means of regularization that mitigates over-fitting. More strategies, e.g., model (i.e., tree) selection that determines the optimal amount of model complexity (Hastie et al., 2009), are worth investigating in the automated design field to cope with the challenge. The GP generalization survey in Agapitos et al. (2019) provides a comprehensive analysis of these strategies. Another challenge is that the tree has been limited to represent a single algorithmic operator. The indirect encoding (Eiben and Smith, 2015; Stanley et al., 2019) may alleviate this challenge. Its generative and developmental nature allows learning and reusing building blocks of the tree, which could help improve the search efficiency and scale up the complexity of the algorithm being represented (Eiben and Smith, 2015).

**Linear array representation with computational primitives:** It uses a linear array of computational primitives to represent a metaheuristic operator (Goldman and Tauritz, 2011; Woodward and Swan, 2012). An example is shown in Figure 3. The primitives execute sequentially along with the array; primitives are associated with indexes and parameters to determine which input variable(s) the primitive should execute on. Similar to the tree representation, the linear array was utilized to represent a single algorithmic operator instead of an entire algorithm (Goldman and Tauritz, 2011; Woodward and Swan, 2012) because of the weak expressiveness of the linear array. Related works include representing the crossover (Goldman and Tauritz, 2011) and mutation (Woodward and Swan, 2012) operators of GAs.

A way of augmenting the expressiveness is using the Push language (Spector, 2001) to map the linear array of primitives to an executable algorithm (Lones, 2019; Kamrath et al., 2020; Lones, 2021). Push is a Turing-complete, stack-based typed language used in GP (Spector, 2001). With Push, the primitives are typed, and each primitive executes upon its corresponding type stack (Lones, 2021), rather than executing sequentially along with the array. This stack-based typed system enables the linear array to be more expressive and

```
[swap(3, 5), merge(1, r, 0.7)]
```

Figure 3: Example of a metaheuristic operator represented by a linear array of computational primitives. The example is derived from Goldman and Tauritz (2011). It represents a crossover operator consisting of a *swap* and *merge* primitives. The indexes and parameters associated with the primitives are in brackets. This operator produces offspring by first swapping the 3rd element of parent 1 and the 5th element of parent 2, then merging the 1st element of parent 1 (by a weight of 0.7) with a random element of parent 2.

ensures the represented algorithm to be syntactically valid (Lones, 2021). The linear array representation with Push was used to represent individual-based local search (Lones, 2019; Kamrath et al., 2020) and population-based (Lones, 2021) algorithms.

Overall, using the design space with computational primitives indicates that the design is from scratch with little human bias. Although unbiasedness makes the design space more difficult to search than biased ones, it provides the design space generalization to different target problems. Furthermore, this kind of design space has the potential to produce innovative algorithms that go beyond human experience and surpass the performance of existing algorithms. This potential is appealing and could attract more researchers and practitioners to study, analyze, interpret, and apply automated design techniques with the advancement of representations and methods for manipulating the representations.

### 3.2 Design Space with Algorithmic Operators

Operators, e.g., the single-point crossover (Eiben et al., 2003), tournament selection (Eiben et al., 2003), greedy randomized construction (Feo and Resende, 1995), and tabu list mechanism (Glover, 1986), are the functional components of an algorithm. They have been used as building blocks to constitute the design space. This design space enables designing metaheuristic algorithms via choosing operators from the space, composing the chosen operators as an algorithm, and configuring endogenous hyperparameters. The ways of representing algorithms over this design space include linear array representation, graph representation, and tree representation in the literature.

**Linear array representation with algorithmic operators:** The linear array representation is the common option in the literature with design space with algorithmic operators. It uses categorical identifiers to index operators and numerals to refer to (conditional) hyperparameter values; an algorithm is subsequently represented by a linear array of the identifiers and numerals (Oltean, 2005; Dioşan and Oltean, 2009; Khud-aBukhsh et al., 2009; Lopez-Ibanez and Stutzle, 2012; van Rijn et al., 2016). The array is normally with a fixed length for easy manipulation. The common practice of executing the array of operators is predefining an algorithm template that maps the linear array to an executable algorithm. For example, in Blot et al. (2019), a local search template was constructed by four components, i.e., selection, exploration, perturbation, and archive; the template organizes the linear array of operators and parameters into a local search algorithm. in Villalón et al. (2022), the PSO template (Shi and Eberhart, 1998) was adopted to map the linear array to a variant of PSO.

Oltean (2005) and Dioşan and Oltean (2009) provided an alternative to the predefined template, in which each operator was associated with a sequence number; the operators executed according to the sequence. This results in unfolded algorithms with serial executions of operators. The Backus Naur Form grammar (Ryan et al., 1998) was introduced as another alternative to the template (Mascia et al., 2013). It has been widely employed (Tavares and Pereira, 2012; Mascia et al., 2014; Pagnozzi and Stützle, 2019; Miranda and Prudêncio, 2020). As shown in Fig. 4, the grammar is formalized as a tuple $(N, T, S, P)$, where $N$ is the non-terminals (operators); $T$ is the terminals (the algorithm's inputs, e.g., parameters); $S \in N$ is called axiom; $P$ is production rules that manage the operators' execution. This grammar ensures the syntactic validity of the represented algorithm by appropriately setting the production rules (Ryan et al., 1998).

**Graph representation with algorithmic operators:** In contrast to the linear array representation that limits the designed algorithms' structure within the predefined template or production rules, the graph is a well-defined format describing arbitrary orderings of a process, resulting in flexible algorithm structures.

```
N =    initialization, crossover, mutation, selection
T =    number cross, prob reset
S =    <algorithm>
P =    the following production rules:
       <initialization>::=   uniform random | Latin hypercube sample.
           <crossover>::=   one-point | n-point(number cross n).
            <mutation>::=   reset(prob reset).
           <selection>::=   tournament | roulette wheel | greedy.
        number cross::=   1 | 2 | 3.
          prob reset::=   0.1 | 0.2 | 0.3.
           <algorithm>::=   <initialization><selection><crossover><mutation><selection> |
                            <initialization><selection><mutation><selection>.
```

(a) Grammar.

```
<algorithm>::= <uniform random initialization><tournament mating selection><one-point
               crossover><reset(prob reset=0.2)><greedy selection>
```

(b) An algorithm instantiated according to the grammar.

Figure 4: Example of grammar for designing genetic algorithms. In (a), the production rule `<initialization>` means that the initialization operator can be either `uniform random` or `Latin hypercube sample`; the meaning of other rules is in the same fashion. According to the `<algorithm>` rule, two elitism GA structures can be represented: one searches by crossover and mutation; the other searches by mutation only. An algorithm instantiated by the rules is shown in (b).

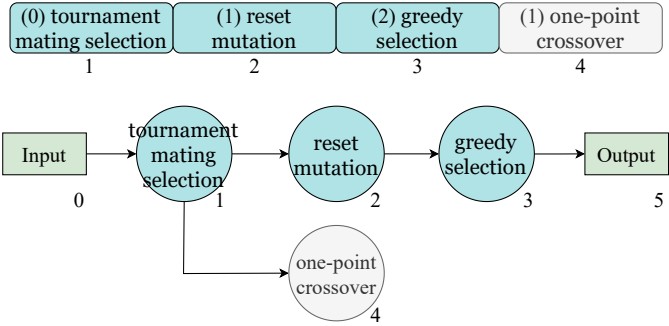

Figure 5: Example of a metaheuristic algorithm represented by a directed acyclic graph. The example is derived from Ryser-Welch et al. (2016). Nodes indexed with $1, 2, 3, 4$ represent algorithmic operators. "(0)" for node 1 means that node 1 connects from node 0; the meaning for other nodes is in the same fashion. The exemplified algorithm consists of one input, one output, and three operators $1, 2, 3$. Operator 4 is inactivated because it is not on the path from the input to the output.

Nodes of the graph refer to operators from the design space; directed edges manage the operators' execution flow. Specifically, the directed acyclic graph from Cartesian GP (Miller, 2011) was employed in Kantschik et al. (1999); Shirakawa and Nagao (2009); Ryser-Welch et al. (2016), in which each node is connected from a previous node or the input in a feed-forward manner; the nodes on the path from the input to the output are activated and constitute the represented algorithm. An example is depicted in Fig. 5.

A more flexible directed cyclic graph representation was proposed in Tisdale et al. (2021), in which nodes were allowed to be connected from multiple previous nodes, and edges were associated with weights and orders to determine i) the computational resources allocated to nodes, ii) the inputs and outputs of nodes, and iii) the starting point of the represented algorithm. These features allow the graph to express various $(\mu/p + \lambda)$ and $(\mu/p, \lambda)$ EA structures. The graph representation in Tisdale et al. (2021) is well-developed to express EA structures but lacks generalization to describe other types of metaheuristics. For this, Zhao et al. (2022; 2023a) proposed a new graph representation, in which i) the directed edges enable sequence structures of operator compositions; ii) each node is open to connecting forward to multiple nodes, allowing branch structures; and iii) cycles realize loop structures. These principles enable the graph to represent different

types of metaheuristic algorithms, e.g., algorithms with unfolded tandem operators, algorithms with inner loops of local search, and algorithms with multiple pathways.

**Tree representation with algorithmic operators:** The binary GP-style tree (Koza, 1994) was also considered to represent an algorithm over operators (Smorodkina and Tauritz, 2007; Rivers and Tauritz, 2013). Terminals of the tree are the algorithm's inputs. Non-terminals of the tree are the algorithm's operators and (conditional) hyperparameters from the design space. This representation is similar to the tree representation with computational primitives in Fig. 2. The difference is that the non-terminals change from computational primitives to operators.

Overall, the design space with existing algorithmic operators is more compact than that with computational primitives. Furthermore, using this design space allows the design inheriting from prior expert knowledge and experimentation. The compact and knowledge-induced space makes the design over the space possibly easier and accessible with fewer computing resources than that over the space with computational primitives. The weakness is that the space may bias the designed algorithms in favor of human-made ones, thereby reducing the flexibility and innovation potential; besides, users may need to select and collect operators to form the space by themselves, which is a new burden.

### 3.3 Usability

The usability of different kinds of design space and representations varies in different algorithm design scenarios. The design space with computational primitives generally requires a large amount of computing resources to fully explore. This design space would be desired for metaheuristic researchers because of its innovation potential. The design space is not the first choice for practitioners and researchers from other communities because it is not easy for these users to interpret the algorithms derived from computational primitives. The tree representation is preferred when using this design space due to the tree's strong expressiveness and maturity along with the advancement of GP.

The design space with existing algorithmic operators can be dense, which makes the space accessible to scenarios with limited computing resources. Furthermore, the space can be formed with good design choices from prior algorithms and the user's domain expertise. Therefore, this space is recommended if such prior algorithms or expertise are available. If the target problem is not complicated and does not require a complex algorithm, the linear array representation may be considered since it is easy to be implemented and manipulated; otherwise, the graph representation is advised because of its strong expressiveness. In addition, the graph representation is easy to understand, which is important for users outside the metaheuristic community.

Both the design space with computational primitives and that with algorithmic operators are mixed and conditional, no matter what representation is used. That is, the space is a mix of discrete primitives/operators and continuous hyperparameter values[2]; the chosen operator conditions what hyperparameters will be involved. The mixed and conditional nature requires specific design strategies (detail in Section 4) to handle. Model-free heuristic methods may be required for directly manipulating the mixed and conditional space; otherwise, the space should be projected to another space with a single type of entities (e.g., discretizing continuous hyperparameter values) and align the representations with different conditional hyperparameters to a unified prototype, in order to leverage model-based and learning-based methods for manipulation.

## 4 Design Strategies

The design strategy is a principled way to generate metaheuristic algorithms over the design space. Based on whether there is statistical inference that guides the design process, current design strategies may be classified into two categories, i.e., model-free strategies and model-based strategies. We first overview the model-free design strategies and discuss their strengths, weaknesses, and challenges in subsection 4.2. We then overview the model-based design strategies with a discussion on its strengths, weaknesses, and challenges in subsection 4.1. Finally, we analyze the usability of the two categories at the end of the section.

---

[2]Some hyperparameters may be discrete.

### 4.1 Model-Free Strategies

Model-free strategies search for desired algorithms in a trial-and-error fashion. They mainly include local search and evolutionary search in the literature.

**Local search:** It mainly refers to the iterative local search (Lourenço et al., 2003) utilized in the ParamILS pipeline (Blot et al., 2019; Tari et al., 2020a; KhudaBukhsh et al., 2009). It is usually incorporated with the design space with algorithmic operators. The key steps are local improvement and global perturbation, in which the former changes one dimension of the algorithm representation; the latter reinitializes the current algorithm once reaching a local optima (Hutter et al., 2009). The local search is often conducted over a dense design space with good choices from prior expert knowledge and experimentation. Therefore, the design strategy itself is often deemphasized.

**Evolutionary search:** Evolutionary search (Eiben et al., 2003) is a more popular design strategy and is widely used in the GP-based algorithm design literature. The algorithm representation determines what specified evolutionary operations can be used. For the tree representation, the sub-tree crossover and muta-tion are common options (Poli et al., 2005a;b; Richter and Tauritz, 2018; Richter et al., 2019; Smorodkina and Tauritz, 2007; Rivers and Tauritz, 2013). That is, for a pair of tree representations, a node is randomly selected from each tree, respectively, and then, the sub-trees starting at the selected nodes are exchanged (crossover); each sub-tree starting at a randomly selected node is reinitialized (mutation) (Eiben et al., 2003). For the linear array representation, various standard evolutionary crossover and mutation are avail-able (Goldman and Tauritz, 2011; Woodward and Swan, 2012; Oltean, 2005; Dioşan and Oltean, 2009; van Rijn et al., 2016; Ryan et al., 1998; Ye et al., 2022a). Evolutionary search is also employed as a meta-learner within the context of meta-learning, such as learning the update rules of evolutionary strategies (ES) (Lange et al., 2023). In particular, Lange et al. (2023) parameterized the update rule of ES by a neural work with the self-attention mechanism to fulfill the invariance in the ordering of ES solutions; the parameters were optimized by evolutionary search, resulting in new ES capable of generalizing to unseen optimization problems.

Evolutionary search is an adaptive system (Holland, 1962) that provides higher-level adaptation across different target problem instances than local search. The adaption is realized by evolving a population of algorithms fitting across the problem instances. The adaption can be further boosted by evolving from easy to difficult instances (if the difficulty is accountable), in which the knowledge found from easy instances can be anchors to enhance the adaption to difficult instances.

An open challenge behind evolutionary design strategies, as well as local search ones, is the lack of principles for handling the conditional design space, i.e., the chosen algorithmic operator conditions what hyperpa-rameters should be involved. The conditional design space essentially makes the algorithm design task to be bi-level. The upper-level searches for the optimal operator composition, and the nested lower-level tunes the hyperparameters within the operators chosen at the upper level. Principle strategies for managing the bi-level search deserve more investigation.

### 4.2 Model-Based Strategies

Model-based strategies use stochastic or statistical models to guide the algorithm design process. They include estimation of distribution, Bayesian optimization, reinforcement learning, and the emerging large language models in the literature.

**Estimation of distribution:** The "estimation of distribution" was the design strategy in irace (López-Ibáñez et al., 2016) [3] It is usually incorporated with the design space with algorithmic operators and the linear array algorithm representation. Its main idea is to build and sample an explicit probabilistic model of the distribution of optimal algorithms over the design space. Specifically, each entity (an algorithmic operator or hyperparameter) of the linear array algorithm representation is assumed to follow a particular distribution; the multinomial distribution and normal distribution are assumed for categorical operators/hyperparameters and numerical hyperparameters, respectively. The probabilistic model of the distribution is formed in an

---

[3]Note that irace did not claim its design strategy as estimation of distribution (López-Ibáñez et al., 2016). We abstract the strategy as an "estimation of distribution" for ease of description.

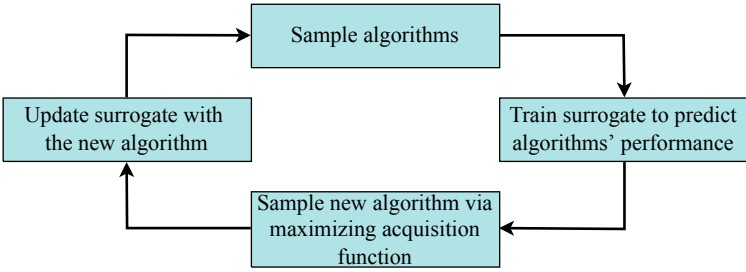

Figure 6: Workflow of Bayesian optimization for algorithm design.

iterative fashion; each iteration consists of three steps: 1) sampling candidate algorithms according to the particular distribution; 2) selecting promising algorithms among the candidate algorithms by means of racing (Maron and Moore, 1997) (detailed in Section 5.3); and 3) updating the distribution by the promising algorithms to approach the optimal distribution. The distribution at the initial iteration can be formed by randomly sampled algorithms or user-specified ones.

The "estimation of distribution" is widely used in the literature (Lopez-Ibanez and Stutzle, 2012; Franzin and Stützle, 2019; Bezerra et al., 2014; 2016; Pagnozzi and Stützle, 2019; Diaz and López-Ibáñez, 2021; Brum et al., 2022) because its fast convergence and the popularity of irace (López-Ibáñez et al., 2016). In its default setting, each entity of the linear array algorithm representation was assumed to follow an independent univariate distribution for ease of manipulation (López-Ibáñez et al., 2016). This indicates the orthogonality among the entities, which is usually not the case and may make the design premature. Learning the covariance of distribution and leveraging other advanced methods, e.g., smoothing the update of the distribution (Wierstra et al., 2014), may mitigate the premature issue.

**Bayesian optimization:** Bayesian optimization is popular in hyperparameter optimization in the automated machine learning field (Hutter et al., 2011; 2019; Lindauer et al., 2022), which has been introduced in metaheuristic algorithm design (Tang et al., 2021; Ye et al., 2022a). It is used to search the design space with algorithmic operators and manipulate the linear array algorithm representations. Its core idea is learning a surrogate that models the mapping from algorithms to their performance on the target problem. The sampled algorithm for training the surrogate is obtained by maximizing the acquisition function. The random forest can be adopted as the surrogate because algorithms should be represented as a mix of numeral parameters and categorical operators/parameters, and random forest supports the mixed training data. The expected improvement (Jones et al., 1998) was utilized as the acquisition function (Hutter et al., 2011) due to its closed-form solution assuming that the designed algorithm's performance follows a normal distribution. The workflow is depicted in Fig. 6.

Bayesian optimization obeys a similar iteration process (i.e., algorithm sampling and model update) with estimation of distribution but with two unique features. First, Bayesian optimization leverages a low-complexity surrogate model to predict the algorithm's performance, which saves computational effort at the cost of potential prediction error. Second, the acquisition function balances the exploration and exploitation in sampling over the design space. The balance is realized by preferring a new sample (algorithm) with high predicted performance (exploitation) and uncertainty (exploration).

Bayesian optimization is acknowledged to perform well on low-dimensional problems. There would be room for improving its scalability to high-dimensional design scenarios. The choice of surrogate models is another open issue. Metaheuristic algorithm design is black-box, in which the landscape of algorithm performance prediction varies when facing different target problems. Apart from the random forest, it is worth having more surrogate choices, e.g., neural networks, to improve the surrogate's accuracy and cope with different design scenarios (regarding target problem types, data richness, computing resources, etc.). Besides, the models can only manipulate fixed-length linear array representations of algorithms. The fixed-length array limits the designed algorithm's structures, subsequently weakening the innovation that can be discovered and leading the model-based strategies only to approach a sub-region of the design space.

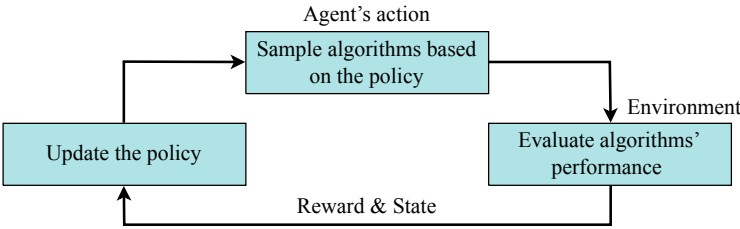

Figure 7: Workflow of reinforcement learning for algorithm design.

**Reinforcement learning:** The Markov decision process was introduced to design metaheuristic algorithms in Adriaensen and Nowé (2016); Meng and Qu (2021). It is executed over the design space with algorithmic operators. The state refers to the operators from the design space (Meng and Qu, 2021) or the algorithms constituted by the operators from the design space (Adriaensen and Nowé, 2016). The transition probability between each pair of states is learned according to the constituted algorithms' performance on the target problem instances. Operators or algorithms with better performance have higher transaction probabilities and are more likely to be chosen. The Markov decision process reveals the intrinsic relationships between each pair of design choices and their contributions to the overall performance (Adriaensen and Nowé, 2016).

Tabular reinforcement learning was employed in Buzdalova et al. (2014); Meng and Qu (2021), in which a tabulation records the reward and penalty of each design choice. Modern reinforcement learning approaches, e.g., deep Q-network (Mnih et al., 2015) and proximal policy optimization (Schulman et al., 2017), were introduced recently (Schuchardt et al., 2019; Sharma et al., 2019; Yi et al., 2023b;a). The action space is the design space with algorithmic operators. The state refers to the information from evaluating the current algorithm on certain problem instances, including features of the instances (e.g., the number and capacity of vehicles when targeting vehicle routing problems (Yi et al., 2023b)), features of the algorithm's solutions to the instances (e.g., solutions' fitness improvement over initial solutions (Yi et al., 2023b)), etc. In particular, the action of Sharma et al. (2019); Sun et al. (2021); Yi et al. (2023b;a) is modifying a certain operator of the current algorithm, resulting in an algorithm with adaptively changed operators during problem-solving. Instead of using a fully connected policy network that only considers the current problem-solving state, Sun et al. (2021) proposed to adopt a recurrent neural network to capture the time-dependent relations from the information till the current state. While the policy is typically learned from scratch, Shala et al. (2020) proposed to use existing ES as a teacher and learn the policy of CMA-ES's step-size adaptation upon the teacher by the guided policy search (Levine and Abbeel, 2014), which accelerated the policy learning process. The general framework of reinforcement learning for metaheuristic algorithm design is given in Fig. 7.

Although reinforcement learning is less used than local and evolutionary design strategies, it is promising in automated algorithm design due to three appealing features. The first is the fine-grained design. Reinforcement learning can learn to design at each time step rather than at each episode (i.e., iteration in the context of iterative local or evolutionary search-based design strategies). This enables building up an algorithm operator-by-operator or primitive-by-primitive, which could be expected to be more efficient than building the whole algorithm at once. The second is the value function. The value function realizes long-term planning of the design with consideration of not only the current state but also future state during a problem-solving. The third is the neural network policy model that restores the design knowledge within the neurons, which enables future reuse to unseen problem instances or transfer to related problems by transfer learning (Taylor and Stone, 2009; Zhu et al., 2023) or continual learning (Khetarpal et al., 2022) techniques.

One concern is that the algorithms derived from the policy are often in linear array representations. Current studies limit the algorithms to the sequential execution of the array. It is worth studying encoding-decoding mapping between the representation manipulated by the agent's action and the executable algorithm to bring innovations in algorithm structures. Another concern is that the reward cannot be calculated in scenarios with fine-grained design from scratch unless all actions have been collected to form a complete algorithm. This arises the sparse reward challenge.

**Large language models:** Language is a complex, intricate system of human expressions (Zhao et al., 2023b). By pre-training Transformer models (Vaswani et al., 2017) over large-scale corpora, large language models (LLMs) are equipped with human knowledge memorization and compositionality capabilities. These capabilities enable LLMs to approximate or even surpass human decisions at an unprecedented level of performance (Zhao et al., 2023b; Mialon et al., 2023). Some works have proposed leveraging LLMs as design strategies to design metaheuristic algorithms. They leverage LLMs through in-context learning, i.e., providing LLMs with a textual prompt and several algorithm demonstrations, LLMs generate algorithms by completing the word sequence of the input text, without training or gradient update (Zhao et al., 2023b). In detail, Romera-Paredes et al. (2024) proposed FunSearch. Given an algorithm template (e.g., a greedy algorithm), FunSearch prompts a pre-trained LLM (Codey (cod, 2023)) to generate a population of programs (code snippets) of the key component (e.g., the priority function of the greedy algorithm); the programs are then evaluated, and feasible ones are sent to a database; the best $k$ programs in the database are selected as new demonstrations in the prompt for the next round of program generation. The above process iterates to find better programs. Precisely, the prompt consists of a description of the target problem, the $k$ demonstrations, and the algorithm template with the key component being empty. Such a prompt aims to let the LLM spot patterns across the best programs and complement the key component in the template based on the patterns. Besides, FunSearch employed an island model that evolves multiple subpopulations of programs, which encourages the discovery of diversified programs. FunSearch discovered new algorithms that improve upon current ones on the cap set and online bin packing problems. This effectiveness is believed to be attributed to the LLM that acts as a source of diverse programs with occasionally interesting ideas rather than using much context about the target problem (Romera-Paredes et al., 2024).

Concurrently, Liu et al. (2024) proposed using the algorithm evolution using large language model (AEL) to design the guided local search algorithm (Voudouris and Tsang, 1999). It adopted an LLM to perform the roles of initialization, crossover, and mutation of an evolutionary process for evolving algorithms. The prompt describes the algorithm design task, parent algorithms, prompt-specific hints, expected output, and other hints. The prompt-specific hints instruct the LLM to perform different roles. Experiments on the traveling salesman problem (TSP) showed that with GPT-3.5-turbo, AEL could evolve elite algorithms in two days with minimal human effort and no model training. Ye et al. (2024) proposed language hyperheuristics (LHHs) that leverage LLMs as high-level automated methods to generate low-level heuristics. In particular, the reflective evolution was presented, in which a generator LLM is for generating heuristics and a reflector LLM is for reflecting hints from the heuristics to guide the next round of generation.

Overall, LLMs bootstrap from rich human knowledge, which could be promising engines for designing algorithms concerning designing from scratch. With LLMs' compositionality over the prior knowledge, an explicit design space is no longer necessary, enabling to discover innovative algorithms. The above works (Romera-Paredes et al., 2024; Liu et al., 2024; Ye et al., 2024) demonstrate such promise. Meanwhile, these works incorporate LLMs with the population-based evolutionary framework. The framework empowers LLMs with diversity and global search capacities (Wu et al., 2024; Chao et al., 2024) and helps mitigate LLMs' hallucination incurred by the lack of fact-checking. A challenge is that a non-trivial prompt engineering is required to reliably extract the knowledge from LLMs. Besides being the core engine of designing algorithms, LLMs could significantly improve human-machine (algorithm design systems) interaction. By building powerful interactive text-based interfaces with LLMs, many complicated decisions affecting the algorithm design process would be simplified. This would also increase the interpretability of the algorithm design process by elaborating on the algorithm design report in textual and graphical forms (Tornede et al., 2023).

## 4.3 Usability

In principle, any search method might be employed to search over the design space to produce algorithms. As algorithm design is black-box, there is not yet a consensus and dominant strategy for designing algorithms. The usability of the design strategies depends on the target problem-solving scenarios. If the design space is formed with a few existing algorithmic operators and there is a predefined algorithm template (e.g., designing a variable neighborhood search algorithm from several neighborhood search operators), the local search, estimation of distribution, and Bayesian optimization could be recommended. Among them, using Bayesian optimization as the design strategy is particularly preferred when targeting expensive problems

Table 1: Summary of design strategies.

| Design Strategies | | Key features |
|---|---|---|
| *Model-Free* | Local Search[1,2]
Evolutionary Search[1,2,3] | [1] With little assumption on the algorithm being designed.
[2] Flexible to be integrated with various algorithm representations.
[3] Exploration over the design space and adaption across problem instances. |
| *Model-Based* | Estimation of Distribution [4,5,6]
Bayesian Optimization [4,5,6,7]
Reinforcement Learning[8,9]
Large Language Models[10,11] | [4] Statistical inference provides interpretation.
[5] Designed algorithms are subject to predefined structure.
[6] Simplified statistics may lead to fast but premature convergence.
[7] Surrogate saves computational effort incurred by algorithm evaluation.
[8] Reveal intrinsic relationships and contributions of design choices.
[9] Fine-grained design, long-term planning, and online design.
[10] Bootstrap from human knowledge, an explicit design space is unnecessary.
[11] User-friendly human-machine interaction. |

in which the computational cost is dominated by function evaluations, since the low-complexity surrogate model can save function evaluations.

When manipulating the design space with computational primitives, evolutionary search, e.g., GP, would be the first choice. Due to the ability to maintain a population of diverse algorithms, evolutionary search is also suitable for scenarios with demands for algorithm portfolios. Reinforcement learning is appealing for scenarios with a large amount of problems from the same or similar domains, in which the policy model enables the reuse and transfer of the knowledge of prior algorithm design to new problems. It is also appealing for online scenarios. That is, learning the design policy offline and then deploying the policy online to generate algorithms during open-ended problem-solving. A summary of the design strategies is given in Table 1.

## 5 Performance Evaluation Strategies

The performance evaluation strategy defines how to measure the performance of the designed algorithms. To conduct performance evaluation, one should first choose a performance metric, then evaluate the performance according to the metric and identify desired algorithms by performance comparison. In particular, reducing the time cost of performance evaluation is vital because performance evaluation normally occupies most of the computational cost of the design process. Hence, we first summarize the performance metrics employed in the literature and discuss their usability in subsection 5.1. Then, we review the performance evaluation and comparison strategies in subsection 5.2. Finally, we report the strategies for reducing the time cost of performance evaluation and discuss their strengths and weakness in subsection 5.3.

### 5.1 Performance Metrics

The metrics include the algorithm's solution quality, running time, anytime performance, and jointly considering multiple metrics in the literature.

**Solution quality:** A commonly used performance metric is the algorithm's solution quality within a fixed computational budget. The solution quality is usually a summarizing value based on the objective function value and constraint satisfaction on the target problem. The black-box optimization benchmarking study in Hansen et al. (2022) constructed solution quality metrics for different types of problems, e.g., single-/multi-objective, constrained, and noisy problems.

**Running time:** It measures the algorithm's running time until the algorithm reaches a performance threshold. The threshold can be the solution quality or a particular solution. The metric is formulated as (Ye et al., 2022a):

$$P(A|i) = r(A, i, \epsilon), \tag{3}$$

where $r(A, i, \epsilon)$ is the running time of algorithm $A$ on target problem instance $i$ till reaching the threshold $\epsilon$. The running time often refers to the number of function evaluations. The wall clock time and CPU

time may also be considered but may bring about unreproducible results due to the difference in hardware, programming language, and coding style.

**Anytime performance:** It measures the algorithm's ability to return high-quality solutions within any computational budget. The common practice to measure the anytime performance is calculating the area under the curve (AUC) of the empirical cumulative distribution function of running time. Given a set of running time points $\{\tau_u | u \in U\}$ and a set of solution quality thresholds $\{\epsilon_v | v \in V\}$, the AUC is calculated as (Ye et al., 2022a):

$$P(A|i) = \frac{\sum_{u \in U} \sum_{v \in V} \langle s(A, i, \tau_u) \geq \epsilon_v \rangle}{|U| * |V|}, \tag{4}$$

where $s(A, i, \tau_u)$ is algorithm $A$'s solution quality on target problem instance $i$ within $\tau_u$ running time; $\langle \cdot \rangle$ returns 1 if $s(A, i, \tau_u) \geq \epsilon_v$, and return 0 otherwise; $|U|$ and $|V|$ return the numbers of elements in sets $U$ and $V$, respectively. A larger AUC value indicates a better algorithm performance.

An alternative way to measure the anytime performance is by the hypervolume indicator (López-Ibánez and Stützle, 2014). That is, the solution quality and running time can be seen as two conflicting objectives, i.e., better solution quality often demands longer running time. Thus, the elements of an algorithm's performance profile (solution quality versus running time) can be seen as a set of Pareto non-dominate points regarding the two objectives. Therefore, the hypervolume value (Zitzler and Thiele, 1999) of the Pareto non-dominate points with respect to a reference point measures the anytime performance. Higher hypervolume values indicate better anytime performance. This method does not require setting a priori solution quality thresholds.

**Multiple-metric:** Simultaneously considering multiple performance metrics is of interest in certain scenarios. in Blot et al. (2016), the solution quality and memory usage were jointly considered for scenarios with memory limitations. In Blot et al. (2017); Tari et al. (2020b); Bezerra et al. (2020), the solution convergence and solution diversity were considered for designing multi-objective algorithms, i.e., multi-objective design of multi-objective algorithms; they used the hypervolume (Zitzler and Thiele, 1999) to measure the algorithm's solution convergence and used the $\Delta$ spread to measure the solution diversity (Deb et al., 2002). By considering multiple performance metrics, the designed algorithm would be more generalized to the target problem-solving scenario than only pursuing a single metric.

**Usability of the metrics:** The solution quality has been employed in most literature. The reason is possibe because presetting the computational budget is easy when targeting the widely-used numerical benchmark problems (where much prior experimentation can be referenced for presetting the budget). The running time and AUC provide a performance guarantee that the designed algorithm could reach the predefined performance threshold within the time budget. The guarantee is significant to practical scenarios. Therefore, the running time and AUC are recommended if the threshold can be defined according to domain expertise or preliminary experimentation. Function evaluations should be the first choice for quantifying the time budget if function evaluations dominate the computational cost. The reason is that it is fair and can provide reproducible results regardless of hardware, programming language, and coding style (Hooker, 1995). The anytime performance is more informative than running time since it comprehensively reveals the performance across the algorithm execution (Hansen et al., 2022). It is preferred in scenarios with demands for anytime behavior, e.g., the designed algorithm deployed in scenarios with variable computational budgets.

## 5.2 Performance Evaluation and Comparison

Due to the stochastic nature, different runs (with different seeds) of a metaheuristic algorithm on the same target problem instance result in inconsistent performance. During the algorithm design process, simply evaluating and comparing candidate algorithms by the performance of a single run is not meaningful. Current strategies of performance evaluation and comparison during the algorithm design process are as follows.

**Conducting descriptive statistics on the performance:** It is straightforward to use descriptive statistics, e.g., mean or median, to evaluate and compare candidate algorithms (Ye et al., 2022a; Junior et al., 2020; Vermetten et al., 2022), but they contain issues. When targeting multiple problem instances, the mean requires the performance values on different instances within the same scale, which is hard to achieve. The

median is robust to outliers but will ignore the majority of the data, e.g., an algorithm may get unqualified performance in 49% of the evaluations, but the median would not capture this.

**Conducting statistical hypothesis test on the performance:** Hypothesis test provides more statistical evidence of to what extent an algorithm performs significantly better than others. The Friedman test and *t*-test have been employed in the literature (López-Ibáñez et al., 2016). The *t*-test assumes samples (i.e., performance metric values) following a normal distribution, while the Friedman test is nonparametric and may be more preferred. In particular, the *p*-value correction in multiple comparisons (i.e., simultaneously comparing more than two algorithms) was suggested to be abandoned in López-Ibáñez et al. (2016), considering that *p*-value correction makes the test conservative, subsequently lowering the discrimination among algorithms.

### 5.3 Reducing Time Cost of Performance Evaluation

The repetitive performance evaluation leads the algorithm design process to be computationally expensive. A number of strategies have been introduced to reduce the time cost of performance evaluation. They reduce the time cost by either reducing the number of performance evaluations or estimating the performance without a full evaluation.

#### 5.3.1 Reducing the Number of Performance Evaluations

This strategy refers to evaluating on a few target problem instances instead of on all instances. Intensification (Hutter et al., 2009; Blot et al., 2016) is a method following this strategy. It evaluates the candidate algorithms' performance instance by instance. The evaluation terminates, and the candidate algorithm is discarded if the algorithm's performance is worse than the incumbent[4] on the current instance. Otherwise, the evaluation continues till it has been evaluated in all instances.

Racing (Maron and Moore, 1997) is also a method that evaluates the performance instance by instance (López-Ibáñez et al., 2016). The difference of racing from intensification (Hutter et al., 2009; Blot et al., 2016) is that there is no incumbent, but all candidate algorithms are evaluated together. In detail, the candidate algorithms' performance is evaluated in the same instances; the candidates whose performance is statistically worse than at least another candidate are discarded. The evaluation continues with the survival candidates on the remaining instances until the number of survivors reaches the minimum, the computational budget is exhausted, or all instances have been evaluated.

Another strategy for saving performance evaluations is only evaluating desired algorithms instead of all candidate algorithms. The dual population of gender-based GA (Ansótegui et al., 2009) follows this strategy. It maintains multiple algorithms during the design process. The algorithms are separated into a competitive and non-competitive population. The competitive population is evaluated on the target problem instances and works on guiding the design towards finding promising algorithms. The non-competitive population works on introducing diversity without evaluation. The two populations interact through genetic recombination.

#### 5.3.2 Estimating the Performance without a Full Evaluation

One such strategy is using a low-complexity surrogate model to estimate the performance instead of exact evaluation (run the algorithm on the target problem). The random forest acted as a surrogate in Bartz-Beielstein et al. (2005); Hutter et al. (2011); Wang et al. (2017); Lindauer et al. (2022). Its input is vectorized representations of the candidate algorithms; the output is the estimated performance values. Thus, it is an end-to-end performance estimation method and is convenient for use.

Another strategy for estimating performance is early stopping the performance evaluation before the computational budget is exhausted. Capping (Hutter et al., 2009; De Souza et al., 2022) is a method following this strategy. There are two capping versions in the literature. The first version was proposed in Hutter et al. (2009) for scenarios with running time as the performance metric. It uses the best algorithm's [5] running

---

[4]Incumbent is the best algorithm found so far in the design process. Its performance in all instances has been evaluated.

[5]The best algorithm can be either the best one found at the current iteration of the design process (trajectory-preserving capping) or the best one found so far of the design process (aggressive capping) (Hutter et al., 2009).

Table 2: Summary of strategies for reducing time cost of algorithm evaluation.

| Methods | | Main ideas |
|---|---|---|
| *Reducing the number of evaluations* | Intensification | Do not evaluate on the next problem instance if performance is worse than the incumbent. |
| | Racing | Do not evaluate on the next problem instance if performance is statistically worse than at least another algorithm. |
| | Dual population | Only evaluate desired algorithms. |
| *Estimating without a full evaluation* | Surrogate | Use low complexity surrogate to approximate performance. |
| | Hutter et al. (2009)'s Capping | Stop evaluating on the current problem instance once running time meets budget. |
| | De Souza et al. (2022)'s Capping | Stop evaluating on the current problem instance once performance profile exceeds bound. |

time as a bound. An algorithm's performance evaluation stops once its running time reaches the bound. Reaching the bound indicates that the algorithm fails to perform better than the best algorithm until the time bound, which means that the algorithm can be discarded.

The second version of capping was proposed in De Souza et al. (2022) and is for scenarios with anytime performance metrics. It also uses a bound to limit the performance evaluation. But the bound is a step function curve that describes previously evaluated algorithms' aggregated performance profile across multiple running time points. For a new algorithm, its performance evaluation stops once its performance profile exceeds the bound at some running time points.

### 5.3.3 Usability

Intensification (Hutter et al., 2009; Blot et al., 2016) and racing (López-Ibáñez et al., 2016) evaluate performance instance by instance. They are available if there are a relatively large number of instances to discriminate algorithms. While intensification and racing can be compatible with any performance metrics, capping (Hutter et al., 2009; De Souza et al., 2022) is specified to scenarios with running time and anytime performance metrics.

Using a surrogate to partially replace performance evaluations is promising due to its end-to-end nature. However, common surrogates (e.g., random forest) require a fixed-length vectorized representation of the algorithm as input. Subsequently, a predefined algorithm template is needed to avoid unnecessarily lengthy vectors. The template limits the designed algorithms to have the same structure. In this regard, it is worth developing algorithm embedding methods to transform algorithms with various structures into compact representations with the same form. Furthermore, currently employed surrogates are regression models estimating algorithms' performance values, which are not always reasonable. For example, suppose there are two algorithms $\{A_1, A_2\}$ with ground truth performance values $\{0.9, 0.91\}$, where 0.9 and 0.91 stand for $A_1$ and $A_2$'s performance, respectively; and there are two estimations $\{0.89, 0.92\}$ and $\{0.91, 0.9\}$. $\{0.91, 0.9\}$ has a smaller mean square error loss than $\{0.89, 0.92\}$ but results in a wrong performance rank. Considering such cases, surrogates that can directly estimate algorithms' performance rank are advised. A summary of the strategies is given in Table 2.

## 6 Target Problems

This section reviews the problems that have been targeted in the literature. Related software is also reported.

### 6.1 Numerical Benchmark Problems

Many works, especially the early ones, employ numerical benchmarks as target problems. Most of them considered continuous optimization problems. The CEC 2005 real-valued parameter optimization benchmarks (Suganthan et al., 2005) were widely used (Poli et al., 2005a;b; Shirakawa and Nagao, 2009; Miranda and Prudêncio, 2015; Lones, 2019; Bogdanova et al., 2019; Kamrath et al., 2020; Miranda and Prudêncio, 2020; Cruz-Duarte et al., 2020; Lones, 2021; Tian et al., 2023). The CEC 2014 real-valued parameter optimization benchmarks (Liang et al., 2013), the black-box optimization benchmark suite (Hansen et al., 2009), and the

comparing continuous optimizers platform used for the GECCO Workshops on Real-Parameter Black-Box Optimization Benchmarking were adopted in Aydın et al. (2017); Villalón et al. (2022); van Rijn et al. (2016); Tisdale et al. (2021), and (Ansótegui et al., 2015; Richter et al., 2019), respectively.

The DTLZ (Deb et al., 2005) and WFG (Huband et al., 2006) test suites acted as target problems in the automated design of multi-objective metaheuristic algorithms (Bezerra et al., 2016; 2020; de Lima and Pozo, 2017). Binary optimization benchmarks were also considered. The $NK$-Landscape benchmarks (Kauffman et al., 1993) were utilized in Goldman and Tauritz (2011); Richter and Tauritz (2018) because the fitness landscape properties can be easily controlled by the $N$ and $K$ parameters. The D-Trap suite (Deb and Goldberg, 1993) was employed in Smorodkina and Tauritz (2007); Goldman and Tauritz (2011); Richter and Tauritz (2018); Rivers and Tauritz (2013).

A common experimental observation of these works is that automated design techniques are able to recur existing algorithms and produce new algorithms that perform better than the existing ones. Due to the expressiveness of various problem landscape features, numerical benchmarks are also important for verifying new automated design techniques and empirically analyzing the techniques, e.g., analyzing the efficiency of different performance metrics (Ye et al., 2022a). A common concern on the numerical benchmarks is that it needs to be clarified to what extent these artificial problems reflect the practice. In this regard, benchmark problems derived from the real world could be considered to stimulate the practicality of automated design techniques.

## 6.2 Practical Problems

Automated design of metaheuristic algorithms has been applied to some practical problem domains, although most of the problems come from experimental simulations. Job shop scheduling (JSS) is a representative practical problem considered in the literature. In Mascia et al. (2013; 2014); Vázquez-Rodríguez and Ochoa (2011); Pagnozzi and Stützle (2019); Bezerra et al. (2014); Franzin and Stützle (2019); Alfaro-Fernández et al. (2020); Sae-Dan et al. (2020); Brum et al. (2022), metaheuristic solvers were automatically designed for JSS with different objectives, e.g., makespan, flowtime, and total tardiness. The bi-objective JSS that simultaneously minimizes makespan and flowtime, the dynamic bi-objective JSS, and the resource-constrained JSS were further considered in Blot et al. (2017; 2019); Nguyen et al. (2014; 2022), respectively.

In Sae-Dan et al. (2020); Tavares and Pereira (2012); Ryser-Welch et al. (2016); Lopez-Ibanez and Stutzle (2012), the automated design was conducted within the ACO template for the TSP. The vehicle routing and nurse rostering problems were adopted as case studies in Qu et al. (2020); Meng and Qu (2021). The propositional satisfiability (SAT) problem with instances from various distributions was targeted in Smorodkina and Tauritz (2007); KhudaBukhsh et al. (2009). The quadratic assignment, bin packing, and imbalanced data classification were considered in Franzin and Stützle (2019); Sae-Dan et al. (2020); Mascia et al. (2014), and (Tari et al., 2020b;a), respectively. The automated design techniques were applied to problems with real-world data in Zhao et al. (2022), in which solvers were automatically designed for the raw material stacking and rack placement problems from the warehouse management department of a biomedical electronics company and were reported to be satisfactory.

The aforementioned works show great potential for automated metaheuristic algorithm design for practical problems. For example, for the JSS problem, new stochastic local search algorithms were designed by the EMILI framework and demonstrated to be superior to the state-of-the-art iterative greedy and local search algorithms when pursuing the makespan, flowtime, and total tardiness objectives (Pagnozzi and Stützle, 2019; Alfaro-Fernández et al., 2020). Multipass heuristics were evolved by GP and outperformed the popular heuristics FIFD, WSPT, WEDD, etc. (Thiruvady et al., 2013; Pinedo, 2012) in resource constraint job scheduling (Nguyen et al., 2022). For the TSP, new metaheuristic solvers were discovered by Cartesian GP from small-scale TSP instances and reported to generalize well to much larger instances (Ryser-Welch et al., 2016). For the SAT problem, stochastic local search algorithms were derived from SATenstein and significantly outperformed previous state-of-the-art algorithms on various benchmark distributions (KhudaBukhsh et al., 2009).

### 6.3 Software

Generally, one can choose a design space, design strategy, and performance evaluation strategy from these reviewed in Sections 3, 4, and 5, respectively, to build a concrete pipeline for automated design, following the workflow in Fig. 1. The established pipelines with open-source code that have been or can be introduced in designing metaheuristic algorithms are reported below.

**irace:** irace (López-Ibáñez et al., 2016) is an iterative version of the racing method (Maron and Moore, 1997). irace for metaheuristic algorithm design uses (i) the design space with existing algorithmic operators and the linear array representation, (ii) "estimation of distribution" as the design strategy, and (iii) racing as the performance evaluation strategy. irace has been applied to design metaheuristic algorithms with given templates (e.g., with templates of ACO (Lopez-Ibanez and Stutzle, 2012), artificial bee colony (Aydın et al., 2017), SA (Franzin and Stützle, 2019), PSO (Villalón et al., 2022)) and design general metaheuristics (Bezerra et al., 2014; 2016; Pagnozzi and Stützle, 2019; Alfaro-Fernández et al., 2020; Bezerra et al., 2020; Sae-Dan et al., 2020; Diaz and López-Ibáñez, 2021; Brum et al., 2022). The `R` implementation of irace is available on its website[6]. A `C++` interface of irace is given in the Paradiseo software (Dreo et al., 2021; Aziz-Alaoui et al., 2021) and is available on Github[7].

**ParamILS:** ParamILS (Hutter et al., 2009), as its name indicated, uses the iterative local search (Lourenço et al., 2003) for parameter configuration. It was employed in algorithm design by representing algorithmic operators as categorical parameters (Blot et al., 2019; Tari et al., 2020a; KhudaBukhsh et al., 2009). Apart from using existing algorithmic operators as design space and iterative local search as design strategy, ParamILS proposes capping as the performance evaluation strategy. A multi-objective ParamILS was developed in Blot et al. (2016) to jointly consider multiple performance metrics. The `Ruby` implementation of ParamILS is available on its website[8].

**SMAC:** SMAC (Hutter et al., 2011; Lindauer et al., 2022) was originated to hyperparameter optimization in automated machine learning and was then employed to configure metaheuristic algorithms (Tang et al., 2021). Its difference from ParamILS is that it uses Bayesian optimization as the design strategy. The intensification and capping can be the performance evaluation strategy alternatively. The `Python` implementation of SMAC is available on Github[9].

**Sparkle:** Sparkle is a recently released `Python` platform for selecting and configuring algorithms (van der Blom et al., 2022). Its current version uses SMAC as the configurator and could be extended to algorithm design. The platform provides easy use for non-experts and supports benchmarking and ablation analysis. The source code is available on its website[10].

## 7 Research Trends

The development of automated metaheuristic algorithm design continues along a number of research threads. Some of the research trends are discussed in this section.

### 7.1 Design Space

**Constructing design space with less user effort:** The design space is critical since it determines what algorithms can be found in principle. Till now, the design space still requires users to manually construct. Guidelines or strategies for constructing the design space are important. One way to provide such guidelines or strategies may be benchmarking, i.e., collecting general design choices for different types of problems, i.e., continuous, discrete, and permutation, and offering maintainable and extendable platforms for users to use and analyze the design choices on particular problems. Another way is automatically constructing the design space. This way is emerging in the neural architecture search field (Radosavovic et al., 2020), in which the

---

[6]https://iridia.ulb.ac.be/irace
[7]https://github.com/jdreo/paradiseo
[8]https://www.cs.ubc.ca/labs/algorithms/Projects/ParamILS
[9]https://github.com/automl/SMAC3
[10]https://bitbucket.org/sparkle-ai/sparkle/src/main/

design space was evolved according to the performance of neural networks sampled from it (Ci et al., 2021) or co-evolved along with the architecture search process (Chen et al., 2021). These works pave a way for design space construction.

**Novel representations:** The GP parse tree and linear array are dominant algorithm representations for design space with computational primitives and design space with algorithmic operators, respectively. For the GP tree, it has been limited to represent a single algorithmic operator. The idea of indirect encoding (Eiben and Smith, 2015; Stanley et al., 2019) may break the limitation. Its generative and developmental nature allows learning and reusing building blocks of the tree, which could improve the search efficiency and scale up the complexity of the algorithm being represented (Eiben and Smith, 2015). For the linear array, it requires a given algorithm template, limiting the novelty in algorithm structures. The graph representation (Tisdale et al., 2021) would be promising because of its expressiveness in representing algorithm flows. An open challenge is that the graph is difficult to be directly manipulated. More studies are desired on the encoding and decoding of graph representation to make it easily manipulated and compatible with different design strategies and performance evaluation strategies.

## 7.2 Design Strategies

**Designing diversified algorithms in parallel and distributed manners:** Designing multiple algorithms in parallel facilitates exploration over the design space (Miikkulainen and Forrest, 2021; Ye et al., 2022b). This arises an issue of how to measure the diversity (similarity) of the algorithms. Ideally, the similarity between algorithms is determined by not only the distance between their representations but also their similarity in terms of algorithmic structures and exhibited behaviors (Xu et al., 2016; Nikfarjam et al., 2022; Xiang et al., 2022). Quantifying algorithms' similarities and subsequently identifying unexplored design space deserves more attention. Moreover, the advances of distributed computing platforms (e.g., Hennessy and Patterson (2011; 2019)) provide opportunities for scaling automated design in hosting more challenging and expensive problems, which also deserves attention.

**Learning-based design strategies:** Learning-based design strategies, e.g., reinforcement learning, offer appealing features, including fine-grained design, long-term planning of the design with the value function, and knowledge reuse with the policy model. These features promise reinforcement learning to generate algorithms to fit-for-purpose in online and open-ended problem-solving scenarios with changing problem characteristics, user requirements, and operating environments (Hart, 2017; Weyns et al., 2021). It is desirable to have more studies on developing advanced policies and value functions for learning-based design strategies.

## 7.3 Performance Evaluation Strategies

**Novel performance comparison methods:** While null hypothesis statistical tests have been the common way of performance comparison during the algorithm design process, recent alternatives have shown great potential in complex scenarios. For example, the Bayesian analysis of Rojas-Delgado et al. (2022) provided an inference of algorithm ranking and quantification of the uncertainty involved in the ranking. Quantifying the uncertainty is important, considering the stochastic nature of metaheuristic algorithms. The study in Yan et al. (2023) proposed cumulative distribution functions with an algorithm-independent binary filter condition to address the "cycle ranking" paradox, i.e., inconsistent ranking of algorithms in multiple comparisons. These techniques are worth integrating within the algorithm design process for a more reliable performance comparison.

**Surrogate-based performance estimators:** The automated design would be time costly when targeting expensive problems. The cost can be saved by evaluating on a part of instances or early stopping the evaluation. The former requires adequate instances to discriminate the performance of different algorithms, while the latter is available when using running time or anytime performance metrics. An alternative is to build a low-complexity surrogate model to estimate the performance. This surrogate estimator is end-to-end, which can be integrated with any performance metrics and is independent of the characteristics of the target problem. There are some challenges to building an accurate surrogate. For example, how to embed the algorithms with various structures to the input of the surrogate, how to estimate algorithms' performance

rank rather than performance metric values, and how different surrogate models are workable for different design scenarios (regarding sample richness, computing resources, etc.).

### 7.4 Experimental and Theoretical Analysis, Practical Applications

Although many works have reported remarkable efficiency of automated design, it is unclear what design space, design strategy, and performance evaluation strategy contribute to that efficiency. Further experimental and theoretical analysis is required to reveal the impact of different kinds of design space and design strategies on different problems. Besides, a majority of existing works apply automated metaheuristic algorithm design to numerical benchmarks and classical simulated problems, e.g., JSS, TSP, and SAT. It is important to go beyond these numerical and simulated problems and contribute more to real-world problem-solving, e.g., incorporating problem-specific elements with the design process, enabling the designed algorithms to handle constraints and to decouple large-scale problem space. Benchmarking, platforms, and user interfaces are also appreciated to promote practical applications.

### 7.5 Intersection with Related Research Fields

Automated design of metaheuristic algorithms is closely related to automated machine learning and meta-learning (Hutter et al., 2019). Advances from the related fields would fertilize the development of the automated design of metaheuristic algorithms, such as multi-fidelity performance evaluation to speed up the design process, model learning to generate surrogates from previous experience and learn policies for reinforcement learning-based design strategies. On the other hand, the automated design of metaheuristic algorithms presents key techniques for automated machine learning and meta-learning. For example, metaheuristic algorithms could be high-level methods for conducting automated machine learning and meta-learning; the GP tree-based representation with computational primitives paves a potential way for automatically programming machine learning algorithms. The interaction and integration of these fields may be a promising thread for boosting innovations.

## 8  Conclusion

This paper surveyed the automated design of metaheuristic algorithms. In the survey, we have abstracted the concept and proposed a taxonomy of automated design of metaheuristic algorithms by formalizing the design process into four parts, i.e., design space, design strategies, performance evaluation strategies, and target problems. We have then reviewed the advances regarding the four parts, respectively. The strengths, weakness, challenges, and usability of these works have been discussed. Finally, some of the research trends have been pointed out. Hopefully, this survey can provide a comprehensive and meticulous understanding of the automated design of metaheuristic algorithms, inspire interest in leveraging automated design techniques to make high-performance algorithms accessible to a broader range of researchers and practitioners, and boost automated design innovations to fuel the pursuit of autonomous and general artificial intelligence.

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
