# OpenReview forum: "Automated Design of Metaheuristic Algorithms: A Survey"
_TMLR — Accepted by TMLR_

### Review · Reviewer_R38i · 2023-11-23

**Summary Of Contributions:**

Contributions
* The authors formalize the auto-design of metaheuristic algorithms into four modules (design space, design strategy, performance evaluation strategy, and target problem).
* Design options for each module are discussed. Each of the first three modules ends with a comment on usability of the options.
* The design space module is highlighted, which is not covered in related surveys.

**Audience:**

Yes

**Claims And Evidence:**

Yes

**Requested Changes:**

[Optional] It could be useful to include a figure for the linear array representation (just like Figure 2 is for tree, and Figure 4 is for graph)

**Strengths And Weaknesses:**

As a survey paper, I find the main strengths are:
* The paper is well structured and well written. While there are many in-depth discussions, I find the writing engaging and easy to follow. The paper should be very accessible to a wide range of audience from different communities (e.g., genetic programming, automated machine learning).
* The role of the design space is highlighted, which is not emphasized in prior surveys. I think this is a great decision, since in practice, the design space has great impact on the search efficiency and the performance of the searched algorithm. The discussion is thoughtfully organized into two parts: space with low-level computational primitives and space with high-level algorithmic operators. For each space type, different representation options are given (tree, linear array, graph) and their pros and cons are clearly discussed.
* The usability sections provide practical recommendation for researchers what options to consider for each module.
* The authors provide valuable insights on why certain methods are appealing for certain cases. (e.g. the discussion of reinforcement learning algorithms in section 4.1)

I don't see particular weaknesses in this survey.

---

> ### Author Response · Authors · 2024-01-19
>
> Dear Reviewer R38i,
>
> Thank you very much for your comments, and we immensely appreciate your recognition of the significance and insight of our paper.
>
> As suggested, we have now depicted a figure for the linear array representation and added the figure in the revised manuscript. Since we are not allowed to upload the figure within the Comment textbox, we describe the figure in text as follows: the figure depicts an example of a metaheuristic operator represented by a linear array of computational primitives as "[swap(3, 5), merge(1, r, 0.7)]". The example is derived from [1]. It represents a crossover operator consisting of a swap and merge primitives. The indexes and parameters associated with the primitives are in brackets. This operator produces offspring by first swapping the 3rd element of parent 1 and the 5th element of parent 2, then merging the 1st element of parent 1 (by a weight of 0.7) with a random element of parent 2.
>
> References:
>
> [1] Brian W Goldman and Daniel R Tauritz. 2011. Self-configuring crossover. In Proceedings of the 13th Annual Conference Companion on Genetic and Evolutionary Computation. Dublin, Ireland, 575–582.

---

> > ### Comment · Reviewer_R38i · 2024-01-19
> > **ACK**
> >
> > Thanks for the update. The change sounds good to me.

---

### Review · Reviewer_FTmA · 2023-12-15

**Summary Of Contributions:**

This paper provides a survey of the automated design of metaheuristic algorithms (black-box optimizers). As automatically designed algorithms can reach and sometimes surpass manually designed ones, a lot of methods have been investigated so far. This paper presents the representative techniques from four aspects: design space, design strategies, performance evaluation strategies, and target problems. The contribution of this paper is to survey and provide the existing techniques for four aspects.

**Audience:**

Yes

**Broader Impact Concerns:**

This paper does not have a concern on the ethical implications.

**Claims And Evidence:**

Yes

**Requested Changes:**

- Some state-of-the-art works are missing. It would be better to include the recent approaches, especially a (meta)-learning-based approach, for the automated design of black-box optimization algorithms. For example, the following papers should be considered to be included in this survey.

Robert Tjarko Lange, Tom Schaul, Yutian Chen, Tom Zahavy, Valentin Dalibard, Chris Lu, Satinder Singh, Sebastian Flennerhag:
Discovering Evolution Strategies via Meta-Black-Box Optimization. ICLR 2023

Mudita Sharma, Alexandros Komninos, Manuel López-Ibáñez, and Dimitar Kazakov. 2019. Deep reinforcement learning based parameter control in differential evolution. In Proceedings of the Genetic and Evolutionary Computation Conference (GECCO '19). Association for Computing Machinery, New York, NY, USA, 709–717. https://doi.org/10.1145/3321707.3321813

J. Sun, X. Liu, T. Bäck and Z. Xu, "Learning Adaptive Differential Evolution Algorithm From Optimization Experiences by Policy Gradient," in IEEE Transactions on Evolutionary Computation, vol. 25, no. 4, pp. 666-680, Aug. 2021, doi: 10.1109/TEVC.2021.3060811.

Shala, G., Biedenkapp, A., Awad, N., Adriaensen, S., Lindauer, M., Hutter, F. (2020). Learning Step-Size Adaptation in CMA-ES. In: Bäck, T., et al. Parallel Problem Solving from Nature – PPSN XVI. PPSN 2020. Lecture Notes in Computer Science, vol 12269. Springer, Cham. https://doi.org/10.1007/978-3-030-58112-1_48

**Strengths And Weaknesses:**

[Strengths]
- The broad picture for the automated design of metaheuristic algorithms is provided, which is useful for people who want to start the research on this topic.
- Surveying the automated design methods in terms of four modules, including design space, design strategies, performance evaluation strategies, and target problems, is somewhat novel.

[Weaknesses]
- Some state-of-the-art works are missing. Please see Requested Changes.
- Learning-based design strategies might be the most interesting approach in the machine learning community. However, the contents regarding them are limited (only in 7.2).

---

> ### Author Response · Authors · 2024-01-19
>
> Dear Reviewer FTmA,
>
> Thank you very much for your comments. **For the missing works**, as suggested, we have now added them in Paragraphs 3 and 7 of Section 4.1 of the revised manuscript. They read as (in italics):
>
> _Evolutionary search is also employed as a meta-learner within the context of meta-learning, such as learning the update rules of evolutionary strategies (ES) (Lange et al., 2022). In particular, Lange et al. (2022) parameterized the update rule of ES by a neural network with the self-attention mechanism to fulfill the invariance in the ordering of ES solutions; the parameters were optimized by evolutionary search, resulting in new ES capable of generalizing to unseen optimization problems._
>
> Modern reinforcement learning approaches, e.g., deep Q-network (Mnih et al., 2015) and proximal policy optimization (Schulman et al, 2017), were introduced recently (Schuchardt et al., 2019; _Sharma et al., 2019_; Yi et al., 2022; 2023). The action space is the design space with algorithmic operators. The state refers to the information from evaluating the current algorithm on certain problem instances, including features of the instances (e.g., the number and capacity of vehicles when targeting vehicle routing problems (Yi et al., 2022)), features of the algorithm’s solutions to the instances (e.g., solutions’ fitness improvement over initial solutions (Yi et al., 2022)), etc. In particular, the action of _Sharma et al. (2019); Sun et al. (2021);_ Yi et al. (2022; 2023) is modifying a certain operator of the current algorithm, resulting in an algorithm with adaptively changed operators during problem-solving. _Instead of using a fully connected policy network that only considers the current problem-solving state, Sun et al. (2021) proposed to adopt a recurrent neural network to capture the time-dependent relations from the information till the current state. While the policy is typically learned from scratch, Shala et al. (2020) proposed to use existing ES as a teacher and learn the policy of CMA- ES’s step-size adaptation upon the teacher by the guided policy search (Levine and Abbeel, 2014), which accelerated the policy learning process._
>
> References (the added works):
>
> Robert Tjarko Lange, Tom Schaul, Yutian Chen, Tom Zahavy, Valentin Dalibard, Chris Lu, Satinder Singh, Sebastian Flennerhag: Discovering Evolution Strategies via Meta-Black-Box Optimization. ICLR 2023
>
> Mudita Sharma, Alexandros Komninos, Manuel López-Ibáñez, and Dimitar Kazakov. 2019. Deep reinforcement learning based parameter control in differential evolution. In Proceedings of the Genetic and Evolutionary Computation Conference (GECCO '19). Association for Computing Machinery, New York, NY, USA, 709–717. https://doi.org/10.1145/3321707.3321813
>
> J. Sun, X. Liu, T. Bäck and Z. Xu, "Learning Adaptive Differential Evolution Algorithm From Optimization Experiences by Policy Gradient," in IEEE Transactions on Evolutionary Computation, vol. 25, no. 4, pp. 666-680, Aug. 2021, doi: 10.1109/TEVC.2021.3060811.
>
> Shala, G., Biedenkapp, A., Awad, N., Adriaensen, S., Lindauer, M., Hutter, F. (2020). Learning Step-Size Adaptation in CMA-ES. In: Bäck, T., et al. Parallel Problem Solving from Nature – PPSN XVI. PPSN 2020. Lecture Notes in Computer Science, vol 12269. Springer, Cham. https://doi.org/10.1007/978-3-030-58112-1_48
>
> **For learning-based design strategies**, reinforcement learning, estimation of distribution, and Bayesian optimization are the learning-based strategies that have been used in the automated metaheuristic algorithm design literature. We elaborated on these methods and analyzed their strengths and weaknesses in Paragraphs 6-9 of Section 4.1, Paragraphs 2-3 of Section 4.2, and Paragraphs 4-6 of Section 4.2 of the manuscript, respectively. We also analyzed their usability in Section 4.3 of the manuscript.
>
> In addition to the learning-based strategies, we believe that the topics of this survey, metaheuristic algorithms and automated algorithm design, are important and would attract broad interests from  the machine learning community. The former, metaheuristic algorithms, are an important niche of artificial intelligence, which have been widely employed in machine learning tasks where the gradients are hard to obtain and where the diversity is worth maintaining (by population-based metaheuristics). The latter, automated algorithm design, as an essential part of automated machine learning, is gaining increasing interest along with the pursuit of autonomous and general artificial intelligence.

---

> > ### Comment · Reviewer_FTmA · 2024-01-21
> >
> > Thank you for the response and for adding the references.
> >
> > By the way, I think that there is room for discussion on whether the learning-based approaches should be categorized into "Model-Free Strategies" (Section 4.1). Because several learning-based approaches contain the machine learning model, such as a neural network, in search algorithms, one possibility is that they might belong to Model-Based Strategies or another category.

---

> > > ### Author Response · Authors · 2024-01-22
> > >
> > > Thank you very much for your comment. The reinforcement-learning-based design strategies in which a neural network is used as the policy model or value function should be categorized into "Model-based Strategies". We have now moved these strategies to the Model-based Strategies part. Thank you.

---

> > > > ### Comment · Reviewer_FTmA · 2024-01-23
> > > >
> > > > Thank you for your consideration. It sounds good.

---

### Review · Reviewer_q9oD · 2024-01-05

**Summary Of Contributions:**

The paper presents a survey on different methodologies for automated design of metaheuristics. The paper also presents a taxonomy for the different components the user of such systems need to be concerned with: design space, design strategy, and performance evaluation.

The survey situates itself in the context of other related surveys, mentioning that previous surveys either focus on other problems (e.g., algorithm selection) or lack information.

**Audience:**

Yes

**Broader Impact Concerns:**

Not applicable.

**Claims And Evidence:**

No

**Requested Changes:**

I would like to see a deep re-write of this paper, where the resulting paper will teach important insights, as opposed to only pointing at different papers. The re-write will have a running example where different representations and designed metaheuristics can be shown. The paper will be more general (e.g., in terms of representations of trees). What can someone who read of these papers conclude that someone who read each of the papers individually wouldn't be able to?

Finally, the mathematical notation will be fixed. I believe there is at least one problem in every equation of the paper.

**Strengths And Weaknesses:**

Strengths

This is a survey of an important topic: automatic design of algorithms. Given the constant increase in computational power, one can imagine that systems will be able to invent novel and powerful algorithms. The survey focuses on metaheuristics, which is an important niche in artificial intelligence and operations research.

The survey covers a large number of papers and can be helpful to readers trying to understand "what is out there" in terms of automated design of metaheuristics. The survey is also thorough in the sense that it covers all the parts involved in the process of designing a system that will design metaheuristics.

Summary of Weaknesses (see detailed comments for more information)

1. The survey only superficially covers some of the topics, such as representation of the search space.
2. The paper doesn't fully explain some of the branches of the literature. Often I missed examples that would help me understand what a group of papers achieved. Some of the topics require longer sections so they can be fully explained. The explanations are shallow to the point that I need to read the papers cited to understand the high-level idea of the paper.
3. I normally expect a survey paper to be insightful, to teach me in a few pages something I would have to spend hours reading the literature to learn. This survey doesn't provide such insights. The closest it gets from providing insights is when the text points out that some of these areas require more research.
4. The mathematical notation of the paper is confusing.

Detailed Comments

The binary tree representation explained in the survey is specific to binary operators. That is, the tree used as an example in Figure 2 is binary because it deals with multiplication, addition, and subtraction. How about a more general representation such as Abstract Syntax Trees? That way, if the language defining the space included ternary operators, the tree would include nodes with three children, to follow the number of "non-terminals" in each rule.

The text mentions that the binary trees present challenges related to overfitting and underfitting. This is a misconception. It is not the representation that presents this challenge, but the space they represent. For example, one could use trees to represent all programs one can write in C++. This space will certainly allow for overfitting as one can write a C++ program that will match exactly whichever data they have. If the language only allows for arithmetic operations, then the system will likely suffer from underfitting, giving that one cannot represent many hypotheses in this space.

An example of the linear array representation would be helpful as I could not understand this representation from the description of the text. Similarly, an example of the grammar shown in Figure 3 would also be helpful. The example in Figure 4 would be more clear if the operations a, f, b, and g were concrete. The paper would probably benefit from having a running example for which we see many different representations of solutions to the example.

Why is it difficult to manipulate a graph? I believe this is another misconception. What the authors meant is that the space the graph represents is too large. This is because it is not difficult to manipulate the graph structure as one can easily define edit operations for graphs.

Some terms are used in the text without explanation. For example, what are phenotypes? What is a bloated space?

The paper poses local search and evolutionary search as two different topics. However, to me, evolutionary search is a type of local search. It often considers many local points, but it is still performing local search, as opposed to enumerative search.

The paper mentions the advantages of Reinforcement Learning as the ability of transferring the value function and policy to other problems. However, it doesn't mention to possible issues with this approach. That is, often the value function and policy will not transfer from one task to the next, even if they are similar.

I don't understand the approaches related to Estimation of Distribution. This is one of the branches of the literature that needs to be better explained. The best the paper does is to point at several papers that use this type of approach.

In Equation 1, the use of the expectation operator $E[⋅]$ is ambiguous, as it is not specified over which variable or set of variables the expectations are being computed. Function $P$ is defined over two parameters, but only one appears in the equation. Also, there is no need to add $A \in S$ as a constraint; one just needs to write $A \in S$ under the argmax operator.

Equation 2 is equally confusing due to the expectations: over which variables are they being computed?

I don't understand the notation $P(A)|i$ in Equation 3. Also, it isn't clear how the minimum between $r$ and $b$ can be meaningful. This is because they represent different quantities.

Equation 5 doesn't make sense to me. How come $r(C) \times N = r(C \times N)$. The value of $C$ isn't defined.

---

> ### Author Response · Authors · 2024-01-19
> **Response to Weaknesses**
>
> Dear Reviewer q9oD,
>
> Thank you very much for your comments. We have carefully studied all the comments and revised the paper according to the comments and suggestions. Below is our response:
>
> **Response to Weakness 1:** As stated in the survey taxonomy of Sections 2.2, survey scope of Section 2.3, and Figure 1 of the manuscript, the survey covers the whole process of automated metaheuristic algorithm design by elaborating on all topics regarding not only the design space (Section 3), but also design strategies (Section 4), performance evaluation strategies (Section 5), and targeted problems (Section 6).
>
> For the representation of the design space, the survey covers all types of representations that have appeared in the automated metaheuristic algorithm design literature, including tree representation, array representation, and graph representation; different variants within each type are elaborated.
>
> **Response to Weakness 2:** Thank you very much for your comments. We have now carefully gone through the manuscript and added fully explainations and examples to the key parts. Please refer to our responses to your Detailed Comments 3, 5, and 8 for the details.
>
> **Response to Weakness 3:** This paper, for the first time, conducts a systematic survey on all essentials of automated design of metaheuristic algorithms:
> - First, the role of design space is highlighted; strengths, weaknesses, and challenges of each type of design space and algorithm representations are analyzed in Paragraphs 3 and 6 of Section 3.1 and Paragraphs 6 and 8 of Section 3.2, respectively; the usability of different types of design space and representations is discussed in Section 3.3.
> - Second, key features, strengths, weaknesses, and challenges of each type of design strategies are analyzed in Paragraphs 4, 5, 8, 9 of Section 4.1 and Paragraphs 3, 6,7 of Section 4.2, respectively; the usability of different design strategies is discussed in Section 4.3.
> - Third, different strategies for saving performance evaluation and their usability are summarized and discussed in Section 5.3.
> - Finally, research trends of this field are highlighted in Section 7.
>
> The above are the valuable insights provided by this survey, which are unavailable by reading separate papers and not given in prior surveys.
>
> **Response to Weakness 4:** Thank you very much for pointing out this. As suggested, we have now revised all mathematical formulations. Please refer to the response to Detailed Comments 9-12 for details.

---

> ### Author Response · Authors · 2024-01-19
> **Responses to Detailed Comments 1-5**
>
> **Response to Detailed Comment 1:** Thank you very much for your comment. The survey only reports binary tree because it is the tree structure that was employed in the automated metaheuristic algorithm design literature. Binary tree is, in fact, the fundamental type of abstract syntax trees and is general to represent different operators [1]. A ternary operator, e.g., “sum(a, b, c)”, can be directly transformed to a binary tree as “sum(sum(a, b), c)”, to name a few.
>
> **Response to Detailed Comment 2:** Thank you for your comments. The underfitting and overfitting of the tree representation is a well-recognized challenge [2]. That is, a very large (small) tree in terms of depth and width results in a very long (short) program, which incorporates too much (little) specific information from the training set, thus overfitting (underfitting) it [3]. Many works have been dedicated to address this challenge by e.g., pruning [4].
> A serious limit in the choices in the design space will of course lead to underfitting. Therefore, maintaining fundamental primitives, e.g., arithmetic, trigonometric, probabilistic and statistic primitives, in the space is necessary, which is a common practice rather than a challenge in the literature.
>
> **Response to Detailed Comment 3:** Thank you very much for your comment. As suggested, we have now added examples of the linear array representation and the grammar; we have also revised the example in Figure 4 by concrete operators. Since we are not allowed to upload figures within the Comment textbox, we describe the example of the linear array representation and that of the grammar in text as follows: The example of the linear array of computational primitives is "[swap(3, 5), merge(1, r, 0.7)]". It is derived from [5], which represents a crossover operator consisting of a swap and merge primitives. The indexes and parameters associated with the primitives are in brackets. This operator produces offspring by first swapping the 3rd element of parent 1 and the 5th element of parent 2, then merging the 1st element of parent 1 (by a weight of 0.7) with a random element of parent 2. The example of grammar is “<algorithm>::= <uniform random initialization><tournament mating selection><one-point crossover><reset(prob reset=0.2)><greedy selection>”. It represents a genetic algorithm, each iteration of which is a serial execution of the tournament mating selection, one-point crossover, reset mutation, and greedy selection.
>
> **Response to Detailed Comment 4:** Thank you for your comment. We agree that graph can be manipulated by heuristic edit operations. However, this indicates that the graph has to be manipulated by model-free search-based methods, e.g., local and evolutionary search. The wide range of model-based design methods, including Bayesian optimization, reinforcement learning, cannot directly manipulate a graph. Normally, a graph embedding should be incorporated with the model-based methods, which complicates the algorithm design task. Therefore, from a generality point of view, graph is not easy to manipulate by a majority of design methods. Compared with acyclic graph, cyclic graph incurs a larger search space because there would be many meaningless cycles that do not connect the output forward to the beginning of the next algorithm iteration. We have now clarified these points in the manuscript.
>
> **Response to Detailed Comment 5:** Thank you very much for your comment. As suggested, we have now revised the text to ensure they are easy to understand. “Phenotype” is a terminology within a main stream of metaheuristic algorithms, genetic algorithms. We now realize that it may not be familiar to the broad machine learning community, so we have now revised the sentence with phenotype as _“An algorithm template is required to insert the operator represented by the tree to an algorithm. The template determines the algorithm structure, i.e., the execution order and logic of the involved operators.”_ We have also replaced “bloated” as “large” to describe that a tree is large in terms of width and depth.
>
> References:
>
> [1] Koza J R. Genetic programming as a means for programming computers by natural selection[J]. Statistics and computing, 1994, 4: 87-112.
>
> [2] Langdon W B, Poli R. Foundations of genetic programming[M]. Springer Science & Business Media, 2013.
>
> [3] O’Neill M. Riccardo Poli, William B. Langdon, Nicholas F. McPhee: A Field Guide to Genetic Programming: Lulu. com, 2008, 250 pp, ISBN 978-1-4092-0073-4[J]. 2009.
>
> [4] Silva S, Costa E. Dynamic limits for bloat control in genetic programming and a review of past and current bloat theories[J]. Genetic Programming and Evolvable Machines, 2009, 10: 141-179.
>
> [5] Brian W Goldman and Daniel R Tauritz. 2011. Self-configuring crossover. In Proceedings of the 13th Annual Conference Companion on Genetic and Evolutionary Computation. Dublin, Ireland, 575–582.

---

> ### Author Response · Authors · 2024-01-19
> **Response to Detailed Comments 6-9**
>
> **Response to Detailed Comment 6:** Thank you very much for your comment. We agree that from the search behavior point of view, local and evolutionary search perform similar. However, as design strategies, they show distinctly different features, benefits, and are suitable for different design scenarios. As stated in Paragraphs 2 and 3 of Section 4.1 and Paragraph 2 of Section 4.3 of the manuscript, local search exploits upon a given algorithm. It is suitable for scenarios where good choices are available from prior expert knowledge and experimentation. In comparison, evolutionary search is an adaptive system [1] that provides higher-level adaptation across different target problem instances than local search by evolving a population of algorithms fitting across the problem instances. It is particularly suitable when an algorithm portfolio is desired. With the above considerations, the survey separates local and evolutionary search as two design strategies.
>
> **Response to Detailed Comment 7:** Thank you very much for your comment. We agree that the policy cannot be directly utilized if the training and test data are not i.i.d. However, within the context of algorithm design for optimization problem-solving, it is a common sense that the instances drawn from a problem are normally follow i.i.d [2].  Therefore, the policy model can be reused for unseen problem instances. For similar problems, the policy model can be reused by transfer learning (e.g., fine tuning) or continual learning techniques. Many work have dedicated on the transfer reinforcement learning [3][4] and continual reinforcement learning [5], which pave the way of reuse prior design knowledge to similar problems. We acknowledge that there are still challenges in transfer and continual learning, e.g., the elasticity and plasticity dilemma. But they are out of the scope of automated metaheuristic algorithm design in this paper, so we did not discuss them.
>
> We have now revised the statement to clarify the point, which reads as _” The third is the neural network policy model that restores the design knowledge within the neurons, which enables future reuse to unseen problem instances or transfer to related problems by transfer learning [3][4] or continual learning [5] techniques.”_
>
> **Response to Detailed Comment 8:** Thank you very much for your comment. As suggested, we have now added details of the estimation of distribution approach, which reads as: _“Its main idea is to build and sample an explicit probabilistic model of the distribution of optimal algorithms over the design space. Specifically, each entity (an algorithmic operator or hyperparameter) of the linear array algorithm representation is assumed to follow a particular distribution; the multinomial distribution and normal distribution are assumed for categorical operators/hyperparameters and numerical hyperparameters, respectively. The probabilistic model of the distribution is formed in an iterative fashion; each iteration consists of three steps: 1) sampling candidate algorithms according to the particular distribution; 2) selecting promising algorithms among the candidate algorithms by means of racing (detailed in Section5.3.1); and 3) updating the distribution by the promising algorithms to approach the optimal distribution. The distribution at the initial iteration can be formed by randomly sampled algorithms or user-specified ones.”_
>
> **Response to Detailed Comment 9:** Thank you for pointing out this. As suggested, we have now revised the equation, which reads as:
>
> $\mathop{\arg\max} _{A\in\mathcal{S}}\ \mathbb{E} _\mathcal{I}[\mathbb{E} _{\mathcal{P}}[P(A|i)]], i\in\mathcal{I}, P\in\mathcal{P}$
>
> where $A$ is the algorithm to be designed; $\mathcal{S}$ is the design space, from where $A$ can be instantiated; $i$ is an instance from the target problem domain $\mathcal{I}$; $P:\mathcal{S}\times\mathcal{I}\to\mathbb{R}$ is a metric that scores the performance of $A$ by a run of $A$ on $i$. Because metaheuristic algorithms conduct stochastic search, we need to estimate the expected performance over $\mathcal{P}$, i.e., multiple runs of $A$ result in multiple $P\in\mathcal{P}$.
>
> References:
>
> [1] Holland J H. Outline for a logical theory of adaptive systems[J]. Journal of the ACM (JACM), 1962, 9(3): 297-314.
>
> [2]Schede E, Brandt J, Tornede A, et al. A survey of methods for automated algorithm configuration[J]. Journal of Artificial Intelligence Research, 2022, 75: 425-487.
>
> [3] Taylor M E, Stone P. Transfer learning for reinforcement learning domains: A survey[J]. Journal of Machine Learning Research, 2009, 10(7).
>
> [4] Zhu Z, Lin K, Jain A K, et al. Transfer learning in deep reinforcement learning: A survey[J]. IEEE Transactions on Pattern Analysis and Machine Intelligence, early access, 2023.
>
> [5] Khetarpal K, Riemer M, Rish I, et al. Towards continual reinforcement learning: A review and perspectives[J]. Journal of Artificial Intelligence Research, 2022, 75: 1401-1476.

---

> ### Author Response · Authors · 2024-01-19
> **Response to Detailed Comment 10-12 and Requested Changes**
>
> **Response to Detailed Comment 10:** As suggested, we have now revised the equation, which reads as:
>
> $\mathop{\arg\max} _{A\in\mathcal{S}} \ \mathbb{E} _{I _{t}} \big [\mathbb{E} _{\mathcal{P}}[P(A|i)] \big ], \ i\in I _{t}\subseteq\mathcal{I}, \forall t\in\{1,2,\cdots,T\}, \ P\in\mathcal{P}$
>
> **Response to Detailed Comment 11:** We have now simplified the formulation to make it clear. It reads as:
>
> Running time measures the algorithm's running time until the algorithm reaches a performance threshold. The threshold can be the solution quality or a particular solution. The metric is formulated as
>
> $P(A|i)=r(A,i,\epsilon),$
>
> where $r(A,i,\epsilon)$ is the running time of algorithm $A$ on target problem instance $i$ till reaching the threshold $\epsilon$.
>
> **Response to Detailed Comment 12:** We have now replaced the equation with text description to clarify the point, which reads as: _”A number of strategies have been introduced to reduce the time cost of performance evaluation. They reduce the time cost by either reducing the number of performance evaluations or estimating the performance without a full evaluation.”_
>
> **Response to Requested Changes:** As suggested, we have carefully gone through the manuscript and rewritten the key parts. Please refer to the responses to the Detailed Comments for details of the main revision.

---

> ### Author Response · Authors · 2024-01-31
>
> Dear Reviewer q9oD,
>
> Thank you again for providing insightful comments that helped us improve our paper, and we hope our responses and updates have addressed your concerns. The Response and Discussion procedure will be due in three days; please let us know if you have any further questions. Thank you.

---

### Comment · Action_Editor_qEqf · 2024-01-19

Dear Reviewers,

Thank you for your great help on reviewing this submission.

The authors have now provided their responses to all reviewers. Please check whether your concerns have been properly addressed or if there are any remaining issues. A direct response to the authors is also appreciated.

Best Regards,

AE

---

### Decision · Action_Editor_qEqf · 2024-02-15

**Recommendation:** Accept with minor revision

**Comment:**

In the initial reviews, reviewer R38i and reviewer FTmA appreciate the contributions and strengths of this survey, and also provide some suggestions (e.g., missing related works and contents, more illustrative figures) for further improvements. These issues have been well addressed during the rebuttal, and they recommend accepting this work. Reviewer q9oD gives a list of detailed and constructive criticism on the weaknesses in the initial review, and the authors provide a thorough response to tackle these concerns. In the official recommendation, reviewer q9oD finds that some of their concerns (especially those on writing) have been addressed, but some deeper feedback (the survey is superficial, lack of insight into existing literature) are overlooked. In addition, this reviewer wants to remind the authors that they can upload an edited version of the paper during the discussion phase, which could be very useful for the rebuttal. Finally, given the potential value of this survey for newcomers in the field, reviewer q9oD leans toward accepting this work.

I read this paper in detail and also believe this work is valuable for anyone who want to work on automated metaheuristic algorithm design. One thing that can be improved is to include a brief discussion on large language model (LLM) for metaheuristic algorithm design. Very recently (and actually after the submission of this survey), some interesting works have been proposed to leverage the power of LLM to automatically design powerful metaheuristic algorithms to tackle different real-world application problems such as [1,2,3]. There are also two survey papers on LLM for evolutionary algorithms from the metaheuristic community [4,5]. Since LLM for algorithm design could be a promising research trend, adding a brief timely discussion on this topic could be very useful for the readers. This discussion can also further strengthen this survey on addressing reviewer FTmA's concern on limited content on learning-based algorithm design. Therefore, I recommend **accept with minor revision** for this work.

In the revised paper, I expect the authors to:

+ Double-check and ensure all the discussions/analyses provided in the rebuttal are well incorporated in the revised paper.

+ Add a brief discussion on large language model for algorithm design.

Given that it is a survey paper, I also recommend a Survey Certification for this work.

[1] Mathematical discoveries from program search with large language models, Nature 2024.

[2] An example of evolutionary computation+ large language model beating human: Design of efficient guided local search, arXiv:2401.02051.

[3] ReEvo: Large Language Models as Hyper-Heuristics with Reflective Evolution, arXiv:2402.01145.

[4] Evolutionary Computation in the Era of Large Language Model: Survey and Roadmap,  arXiv:2401.10034.

[5] A match made in consistency heaven: when large language models meet evolutionary algorithms, arXiv:2401.10510.

**Audience:**

All reviewers believe some individuals in TMLR's audience could be interested in the findings of this paper.

**Claims And Evidence:**

This work conducted a comprehensive survey on the automated design of metaheuristic algorithms, of which related works are systematically reviewed and discussed from four aspects: design space, design strategies, performance evaluation strategies, and target problems. Some research trends for these four aspects have also been disscused at the end of the survey. All reviewers find this work is valuable and useful for researchers/practitioners who want to work on automated metaheuristic design, and believe the claims in this survey are well supported. I totally agree.

---

> ### Author Response · Authors · 2024-02-21
>
> Dear Action Editor,
>
> We immensely appreciate your recognition and valuable comments for the paper. As suggested, we have double-checked the manuscript to ensure that all the discussions/analyses provided in the rebuttal have been incorporated into the camera-ready version. We have also added the related works that you suggested and discussions on large language models for algorithm design in the last three paragraphs of Section 4.2 of the camera-ready version of the paper. Thank you.